# Age-Related Trajectories of Brain Structure–Function Coupling in Female Roller Derby Athletes

**DOI:** 10.3390/brainsci12010022

**Published:** 2021-12-25

**Authors:** Derek C. Monroe, Samantha L. DuBois, Christopher K. Rhea, Donna M. Duffy

**Affiliations:** Department of Kinesiology, University of North Carolina at Greensboro, Greensboro, NC 27402, USA; sldubois@uncg.edu (S.L.D.); ckrhea@uncg.edu (C.K.R.); dmduffy@uncg.edu (D.M.D.)

**Keywords:** female athletes, brain aging, mTBI, collision sports, structure–function coupling, graph signal processing

## Abstract

Contact and collision sports are believed to accelerate brain aging. Postmortem studies of the human brain have implicated tau deposition in and around the perivascular space as a biomarker of an as yet poorly understood neurodegenerative process. Relatively little is known about the effects that collision sport participation has on the age-related trajectories of macroscale brain structure and function, particularly in female athletes. Diffusion MRI and resting-state functional MRI were obtained from female collision sport athletes (*n* = 19 roller derby (RD) players; 23–45 years old) and female control participants (*n* = 14; 20–49 years old) to quantify structural coupling (S_C_) and decoupling (S_D_). The novel and interesting finding is that RD athletes, but not controls, exhibited increasing S_C_ with age in two association networks: the frontoparietal network, important for cognitive control, and default-mode network, a task-negative network (permuted *p* = 0.0006). Age-related increases in S_C_ were also observed in sensorimotor networks (RD, controls) and age-related increases in S_D_ were observed in association networks (controls) (permuted *p* ≤ 0.0001). These distinct patterns suggest that competing in RD results in compressed neuronal timescales in critical networks as a function of age and encourages the broader study of female athlete brains across the lifespan.

## 1. Introduction

Physical activity and exercise promote healthy brain network organization across the lifespan [1,2,3,4]. Sports are commonly touted as an enjoyable way to obtain these benefits, yet sports carrying a risk of incidental contacts between a player and other people or objects (contact sports) and sports that involve purposeful collisions (collision sports) can result in mechanical loading of the athlete’s head [5,6,7], which may accelerate cognitive aging [8,9,10,11]. One recent retrospective cohort study highlighted the cost–benefit paradox of participating in these sports, reporting that former professional soccer players exhibited lower rates of all-cause mortality than matched controls but an elevated risk of neurodegenerative disease [12]. The pathogenic mechanisms linking exposure and neurodegeneration are as yet poorly understood. Currently, these processes are only detectable by postmortem assays of phosphorylated tau in neurons, astrocytes, and the perivascular space around small vessels at the depths of the cortical sulci [13]. Attempts to relate antemortem brain imaging [14] or biofluid biomarkers [15] to clinical measures have yielded mixed results.

Ultimately, complex behaviors and cognition arise from brain structural and functional organization, which is disturbed in a dose-dependent manner by a single season of head mechanical loading [16,17,18,19], after many years of exposure in retired professional athletes [20,21,22,23], and in patients with various tauopathies [24,25]. It is increasingly understood that relationship between structure and function, so-called structure–function (de)coupling, is more strongly associated with age than either constituent measure [26,27] and is disrupted in neurodegenerative diseases [28,29]. Zhang and colleagues [30] recently reported opposing relationships between global structure–function (de)coupling and amyloid beta deposition between patients with Alzheimer’s disease and patients with mild cognitive impairment, suggesting that conversion may be due to the effects of protein accumulation on brain network organization but not by protein burden per se. However, as the authors note, structure–function (de)coupling is not a globally homogenous brain feature. Regional patterns of structure–function (de)coupling produce a distinct ‘sensory-fugal’ gradient, from tight BOLD alignment with the underlying white matter skeleton in sensorimotor networks to liberal alignment and decoupling in transmodal networks and the association cortex [31]. These patterns are associated with cognition [32], and differences in age-related patterns of (de)coupling between collision sport athletes and controls may provide insight into the effects that head mechanical loading has on intrinsic functional dynamics and brain aging.

The study of age-related patterns of structure–function (de)coupling in collision-sport athletes also has potential to reveal patterns that may be clinically meaningful for other populations at risk for sustaining head mechanical loading, including military service members [33] and victims of domestic abuse [34]. However, neuroimaging studies of athletes have historically focused on college-age young adults, with small variance in age, or retired professional athletes, whose experiences and exposure are unlikely to generalize to broader populations. On the other hand, it is estimated that 30–50% of American adults 18–49 years old participate in sports and collision sports (e.g., soccer, football, basketball) are some of the most popular [35]. To the authors knowledge, no report of brain structure or function in this population of athletes has been previously published. Moreover, despite evidence of sex differences in brain aging [36,37], few neuroimaging studies are designed to exclusively study the brains of female contact or collision sport athletes [38].

In the current study, we sought to address these gaps by comparing trajectories of brain aging between female roller derby (RD) athletes and non-roller derby athletes (controls). RD is an understudied, collision sport that was developed as a combination of ice hockey, football, and wrestling and is now played primarily by female amateur athletes [39]. A 2018 survey of 1395 current female RD athletes revealed that 48.7% had sustained one injury in the last year and 35% of those athletes reported a second injury during the same time period [40]. While only 8% (*n* = 111) of the injuries in that study were diagnosed concussions, a smaller study of female RD athletes reported that 58% of respondents reported concussive symptoms in the previous year [41]. Specifically, we tested a hypothesis that female RD athletes would exhibit a different age-related trajectory of brain structure–function (de)coupling compared to females who were not competing in contact or collision sports.

## 2. Materials and Methods

### 2.1. Participants

Experimental Dataset: Nineteen female athletes currently active in a competitive roller derby league were recruited as participants. The roller derby (RD) athletes were 23–45 years old (mean = 32.1 years) and were not diagnosed with a concussion at the time brain imaging was performed. No other data were collected on these participants. Since the purpose of this study was to characterize brain aging in female collision sport athletes, a convenience sample of female control participants (i) not currently engaged in contact or collision sports and (ii) not diagnosed with a concussion in the past three years (*n* = 15) were also recruited to participate. Due to the nascent nature of research in this area, and to optimize sample size, no other exclusionary criteria were used and there was not an attempt to rigorously match participants on age. One control participant’s functional MRI data were deemed unusable and thus that participant was removed from further analysis. The remaining control participants (*n* = 14) were 20–49 years old (mean = 24.6 years) and also self-reported their history of sport participation prior to MRI (Table 1). This study was approved by the Institutional Review Board of the University of North Carolina at Greensboro and written informed consent was obtained from each participant prior to study participation.

### 2.2. MRI Data Acquisition and Preprocessing

Brain imaging was performed using a Siemens Tim Trio 3T system (Siemens, Germany) and a 16-channel head coil. Foam padding was used to minimize the participants’ head motion and all participants wore foam earplugs and headphones to attenuate scanner noise. Before the scan, the participants were instructed to keep their eyes closed, stay awake, and move as little as possible during data acquisition. High-resolution, three-dimensional anatomical images were obtained over 5 minutes using a T1-weighted magnetization-prepared rapidly acquired gradient-echo (MPRAGE) sequence (TR/TE = 1950/4.18 ms; slice thickness = 1 mm; gap = 0 mm; FOV = 256 × 224 mm^2^). An echo-planar imaging (EPI) sequence (TR/TE = 3000/28 ms; flip angle = 73°; slice thickness = 3.3 mm; in-plane resolution: 3.3 × 3.3 mm, 49 slices; matrix = 64 × 64) was used to obtain four-dimensional functional images of the blood oxygen-level-dependent (BOLD) signal at rest over 5 minutes. Diffusion MRI (dMRI) was collected over 12 min using a spin-echo single-shot EPI sequence (TR/TE = 9000/94 ms; 72 directions with b = 0 and 1300 s/mm^2^; flip angle = 90°; slice thickness = 2.7 mm; matrix = 128 × 84).

Structural (T1w) and rs-fMRI data from RD and control participants were preprocessed a containerized version of fMRIPrep version 20.1.2 [42] (RRID:SCR_016216). Boilerplate language is included in the Appendix A. Briefly, brain surfaces were reconstructed from skull-stripped and non-uniformity corrected T1-weighted images using FreeSurfer v6.0.1 [43] (RRID:SCR_001847). Individual reconstructions were spatially normalized to the ICBM 152 Nonlinear Asymmetrical template [44] (RRID:SCR_008796) using the antsRegistration tool [45] (ANTs v2.1.0 RRID:SCR_004757). T2-weighted images were slice-time and motion corrected and co-registered in T1w (native) space using boundary-based registration with six degrees of freedom [46], sampled into high-resolution (164k) fsaverage space as an intermediate step before being re-sampled to 32k fsLR space. fsLR space is a standard mesh comprised of greyordinates represented by 32k vertices (per hemisphere) in cortical surface reconstructions and 28k voxels in subcortical volume. This means that each individual’s resting-state ‘run’ is represented as a 91k greyordinate x 106 TR dense timeseries.

Three region-specific global signals, six head motion parameters (and their respective temporal derivatives and quadratic terms), and framewise displacement (FD) [47] and DVARS [48] were computed from the preprocessed T2-weighted timeseries. Head motion was low for the entire sample (mean FD < 0.5 mm; Max FD < 0.9 mm; Table 1) and did not differ between groups (t(32) = 1.75, *p* = 0.09, Cohen’s d = 0.113). For each participant component-based noise correction was performed using temporal (tCompCor) and anatomical (aCompCor; CSF and WM) physiological noise regressors from an eroded brain mask [49]. The components (CSF, WM, combined, or temporal) that were sufficient to explain 50 percent of variance were retained as confounds. All confounds were regressed out in a single step using ciftify [50], and the resulting mean BOLD timeseries was extracted as the mean signal from 360 cortical areas according to the Cole-Anticevic Brain Network Parcellation [51,52], representing 12 intrinsic connectivity networks: a task-negative network (default), sensorimotor networks (somatomotor, auditory, visual), higher-order associative networks (orbito-affective, frontoparietal, cingulo-opercular, dorsal attention, language), and two novel multimodal networks (ventral, posterior).

Preprocessing of diffusion MRI data was performed using QSIPrep 0.13.0RC [53] and boilerplate language is included in the Appendix A. Briefly, denoising was performed using MP-PCA with a 5-voxel window [54], Gibbs unringing was performed [55], and images were corrected for magnetic field inhomogeneity [56]. Head motion was corrected using the SHORELine method [53,57], wherein each b > 0 image is predicted as a target based on all other b > 0 images. This image was registered to its predicted target and the vector rotation was used to transform the entire set of images. The DWI timeseries were resampled to ACPC, generating a preprocessed DWI run in ACPC space with 2 mm isotropic voxels. No outlier scans were detected based on corrected neighboring DWI correlation values (i.e., all scans were within 3 median absolute deviations).

### 2.3. Structure–Function Coupling

We leveraged recent advances in graph signal processing to characterize the alignment and misalignment (coupling and decoupling) of the observed BOLD timeseries relative to what would be predicted by the structural connectome [58,59,60]. This is different than the more commonly used correlational approach, which defines structure–function (de)coupling based on the strength of the relationship between structural and functional connectivity between one area and all other areas (i.e., the whole-brain weighted degree of each area). All post-processing analytic steps are visualized in Figure 1.

First, preprocessed diffusion data were imported into DSI Studio (http://dsi-studio.labsolver.org (accessed on 13 March 2021)) and reconstructed in template space (R^2^ ≥ 0.78) using q-space diffeomorphic reconstruction [61]. Restricted diffusion imaging [62] was performed to obtain spin distribution functions [63]. One million tracts were reconstructed using a deterministic fiber tracking algorithm based on quantitative anisotropy [64] with parameter saturation, atlas-based track recognition (HCP-1065) [65], and topology-informed pruning to improve rigor and reproducibility [66]. Tracts shorter than 10 mm or longer than 400 mm were discarded. A random angular threshold (15 degrees to 90 degrees) and random step size (0.5 voxel to 1.5 voxels) were used. Fiber propagation direction was averaged with a randomly selected percentage (0% to 95%) of the previous direction to smooth fiber trajectories. Using the same multimodal parcellation to which the functional data were aligned, individual structural connectivity matrices (360 × 360) were defined as the total number of reconstructed tracts connecting areal pairs normalized by the median fiber length. In other words, each individual’s brain is expressed as a network of 360 *nodes* wherein the *edge* between any pair of nodes represents the normalized size of the white matter tracts between them.

Second, a graph Fourier transform was performed such that each individual’s SC matrix was converted from a weighted, symmetric adjacency matrix (*A*) to a symmetric normalized graph Laplacian (*L*) and then subjected to an eigenvector decomposition (*LU* = *UA*). This yielded low-frequency eigenmodes (smaller Laplacian eigenvalues) and high-frequency eigenmodes (larger Laplacian eigenvalues). Third, brain activity was decomposed at each timepoint (or TR) by multiplying the preprocessed functional (BOLD) signal by the greatest Laplacian eigenvectors (to represent decoupled structure–function) and the lowest Laplacian eigenvectors (to represent coupled structure and function). Graph signal filtering was performed based on a median split of the energy spectral density of the decomposed BOLD timeseries. Analyzing the alignment between the BOLD timeseries at each time point (TR) allowed us to determine where and to what extent brain function was coupled or decoupled with white matter anatomy. Thus, the normalized mean of the high-pass filtered, eigenvector-weighted timeseries at each node was interpreted to represent the degree of structural decoupling (S_D_) and the low-pass filtered eigenvector-weighted timeseries at each node was interpreted to represent the degree of structural coupling (S_C_) [67]. The latter steps were performed using modified, publicly available code. (https://github.com/gpreti/GSP_StructuralDecouplingIndex (accessed on 16 March 2021).)

### 2.4. Statistical Analysis

Behavior partial least squares (PLS) correlation analysis was used to test for a joint pattern of areal S_C_ and S_D_ that predicted age and differed between RD and controls [68,69]. The PLS approach was deemed preferrable to traditional, parametric statistical testing since (i) permutation and bootstrapping statistics do not make assumptions about the distribution or normality of the data and (ii) the influence of outliers is mitigated by bootstrap resampling. Analyses were performed on both areal S_C_ and S_D_ because, just as a BOLD timeseries can exhibit low- and high-frequency components, so too can an area exhibit both coupled and decoupled components [67].

A group S_C_ matrix (33 rows × 360 columns) was created by stacking individual S_C_ vectors for controls (14 rows) and RD athletes (19 rows). A S_D_ matrix was created in the same manner. Both matrices were stacked (as ‘within-subject’ conditions nested in groups) to create a 66 row x 360 column matrix as a ‘brain’ input to the PLS. Participants’ ages (years) were stacked similarly to create an ‘age’ input matrix (66 rows × 1 column).

PLS then decomposes the brain-age correlation matrix into orthogonal latent variables, each representing a correlation between some pattern of structural (de)coupling, represented on an individual level as a ‘brainscore’ and age. In total, 10,000 permutations of the multivariate pattern were used to assess the statistical significance of the latent variables, wherein the *p*-value is equivalent to the number of permutations resulting in a singular value greater than what was observed divided by the number of permutations. In other words, only latent variables with fewer than 500 permutations returning greater singular values than what was observed were considered statistically significant (*p* < 0.05). The stability (or reliability) of the contribution of each ‘brain’ feature (S_D_, S_C_) to the latent variable was assessed by bootstrap resampling over 1000 iterations. The resulting distribution was used to compute a standard error, which in turn was used to (i) create a 95% confidence interval around the correlation for each feature and (ii) compute a bootstrap ratio (BSR), essentially a z-score, as each areal feature’s weight over its estimated standard error [68]. Highly positive BSRs are interpreted to mean that (de)coupling contributed reliably to the brain-age relationship identified by the latent variable. Highly negative BSRs to mean that (de)coupling contributed reliably to the opposite brain-age relationship represented by the latent variable. BSRs > 2.58 (corresponding to a 99% confidence interval), representing areas contributing the most to the observed relationship, were averaged within their respective networks to describe which networks contributed, and characterize the direction (coupling, decoupling) of their contribution, to the observed brain-age relationship. In a confirmatory step, we tested for a relationship between the observed patterns and head motion by computing the Pearson correlation coefficients between brain scores for each metric (S_C_ and S_D_) within each LV. Small and non-significant associations were interpreted as evidence that individual differences in head motion were not driving the observed patterns of (de)coupling.

Finally, we considered the possibility that any differences in trajectory might actually be due to non-linear aging trajectories [70], which the PLS is not able to test, and thus be explained by differences in ages between the groups. To address this, we fit two multiple linear regression models (*fitglm* function) for each cortical area:$$y=1+{\;\mathrm{AGE}}_{\mathrm{CEN}}+{\mathrm{AGE}}_{\mathrm{CEN}}^{2}+e$$

In other words, we attempted to predict either structural coupling (S_C_) or decoupling (S_D_) (y) using participant age (centered on the sample mean; AGE_CEN_) and participant age centered and squared (AGE_CEN_^2^). The Benjamini–Hochberg procedure was used to correct for multiple comparisons based on the false discovery rate [71]. The absence of any significant quadratic effect (*p*-adjusted < 0.05) would lend strength to conclusions about group differences, whereas a statistically significant quadratic effect would require interpretation in the context of the data-driven PLS results. Sensitivity analysis (G*Power, 3.1.9.6) revealed power (α = 0.05, 1 − β = 0.80) to detect a medium-to-large effect (Cohen’s f^2^ = 0.25) of two ‘age’ variables in jointly predicting structural (de)coupling in each area.

## 3. Results

The PLS analysis testing the primary hypothesis of different age-related trajectories of structural coupling (S_C_) and decoupling (S_D_) between RD athletes and controls revealed two latent variables (LVs). LV1 (*p* < 0.0001) revealed a set of brain areas for which S_C_ increased and S_D_ decreased with age across both groups (Figure 2a,b). Figure 2b illustrates that, although the correlations between S_C_, S_D_, and age were statistically significant, the pattern revealed by LV1 primarily represents age-related changes in controls. This relationship was primarily attributed to areas comprising sensorimotor (secondary visual, auditory, language, ventral multimodal) networks and the orbitoaffective network (Figure 2c,d). The opposite pattern (increased S_D_ with age in controls) was driven by areas comprising associative networks (i.e., frontoparietal, cinguloopercular, dorsal attention, and default networks).

LV2 (*p* = 0.0006) revealed brain areas for which S_C_ increased with age in RD athletes and S_D_ increased with age in controls (Figure 3a,b). Figure 3b illustrates that, although the correlations were again statistically significant for both groups, the pattern revealed by LV2 primarily represents age-related changes in RD athletes. Similar to LV1, this pattern was driven by areas comprising sensorimotor networks (secondary visual, language, auditory) and increasing S_D_ in the cinguloopercular network. However, contrasting with the patterns in LV1, LV2 also revealed increasing S_C_ in frontoparietal and default networks with increasing age in RD athletes (Figure 3c,d). The latent patterns revealed by PLS were not associated with head motion (Pearson r < |0.12|, *p* > 0.503).

Multiple linear regression models revealed that no cortical area exhibited a quadratic relationship between age and S_C_ (FDR > 0.068) or between age and S_D_ (FDR > 0.70).

## 4. Discussion

Compared to male athletes, relatively little is known about brain structure and function in female athletes, particularly post-collegiate female athletes competing in collision sports. Data collected to fill this gap are especially important for understanding the effects, both healthful and deleterious, of sport participation on brain aging. To this end, we hypothesized that roller derby (RD) athletes and controls would exhibit different age-related trajectories of brain structural coupling (S_C_) and decoupling (S_D_), representing the degree of alignment and misalignment, respectively, between the areal BOLD signal and the underlying white matter skeleton.

Our novel finding was the group differences in age-related (de)coupling in regions comprising the frontoparietal and default networks; in contrast to females not engaged in RD, female RD athletes exhibited age-related increased S_C_ in regions comprising the frontoparietal and default networks. Greater S_C_ could be interpreted to represent timescale shortening, or faster neuronal dynamics, in the frontoparietal and default networks in RD athletes. Aging brains appear to exhibit more global structure–function decoupling [27,72], particularly in subcortical and cerebellar areas and in the frontoparietal and attentional cortical networks [32]. Age-related shifts toward coupling in transmodal networks have also been interpreted to represent healthy aging in older samples than those analyzed here [73], and they form the basis of ‘posterior–anterior shift’ [74] and ‘compensation-related utilization’ [75] models of brain aging. Therefore, it is possible that the pattern we observed does not represent aberrant physiology per se, but instead represents accelerated brain aging: a pattern normally observed in older adults but characterized in young and middle-aged RD athletes in the current study. On the other hand, it is possible to interpret greater S_C_ in the frontoparietal and default networks as representing declining BOLD variability, which recently correlated with declining cognition in a large heterogenous sample of middle-age-to-older adults (43–89 years, *n* = 422) [76], or as the desegregation of the frontoparietal and default networks from the somoto-sensory networks. Although network desegregation is a common feature of aging [77,78], loss of differentiation in the frontoparietal and default networks may represent network-specific decreases in oxygen metabolism [79]. Focal hypometabolism has also been observed in patients with Alzheimer’s disease [80,81] and individuals with a history of concussive exposure [82,83]. Substantial work in the basic science of concussion suggests a time course of cerebral dysmetabolic regulation that begins with acute hypermetabolism, and an insufficient energy supply to meet the demand, giving way to a prolonged period of hypometabolism [84]. Thus, it is possible that increasing structure–function coupling in RD athletes may represent long-term effects of chronic exposure to head mechanical loading. A complete medical history and measurements of head mechanical loading, using head- or mouth-mounted telemetry, would be useful for testing this postulate.

We observed similar patterns of age-related increased S_C_ in areas comprising secondary visual, language, and auditory networks among both RD and control athletes. Control females also exhibited increasing S_D_ with age in regions comprising several transmodal association networks: cinguloopercular, default, frontoparietal, and dorsal attention. These patterns of (de)coupling are consistent with those reported by Preti and DeVille (2019) from a large sample of similarly aged (male and female) participants from the Human Connectome Project Young Adult dataset [67]. As noted by those authors, greater S_C_ is consistent with the shorter timescales observed in firing rates of primary sensory neurons, necessary for the quick transmission of sensory inputs, whereas greater S_D_ likely arises from the slower, tonic firing of neurons in association networks to guide higher-order cognition, such as working memory and decision making [73,85,86]. These patterns are also consistent with patterns of myelin maturation across the age range represented in the current study (20–49 years) [87]. A different analysis of that same dataset (*n* = 420 unrelated males and females aged 22–37 years) reported the most pronounced increases in age-related structure–function decoupling in frontoparietal and attentional cortical networks [32]. Those authors did not report the same age-related increases in coupling in sensorimotor networks, but this may be due to their use of a whole-brain correlational approach—characterizing coupling based on structural and functional connectivity of one area to all other areas—which has different implications for network-based effects than the graph signal processing approach employed in the current study. However, compared to other larger studies correlating brain organization and age, which have not accounted for sport participation, the age range in the current study is relatively narrow, and it is plausible that the study of older athletes (aged 50+ years) would result in a different trajectory.

Our findings are not without limitations. First, participants in our control group were not age-matched with athletes from the RD group, which resulted in a younger control group. Group age differences are partially addressed by the multiple linear regression analysis, which did not support quadratic patterns of age-related (de)coupling across the entire sample. This supports an interpretation that patterns revealed by the PLS analysis represent group differences related to current sport participation rather than artifacts related to these differences in age. Second, roughly one-third of the control participants had previously played soccer, a ‘contact’ (but not a ‘collision’) sport, which, when played at the professional level, appears to increase the risk of neurodegenerative disease [12,88]. If the risks for amateur soccer players are similar, then it is possible that the patterns we observed in LV1 are confounded by the chronic effects of soccer competition. However, this would also lead us to speculate that the patterns observed in RD (LV2) are driven by the acute effects of participation in RD rather than the chronic effects of exposure. Future studies designed to relate brain structure and function to head mechanical loading (e.g., using instrumented mouthguards) in RD athletes is warranted. Third, the lack of behavioral data in the current study also makes it difficult to contextualize the findings. Including a battery of standard neurocognitive tests would clarify whether these patterns represent unhealthy brain aging or not.

Notwithstanding these limitations, the patterns we observed in the current study suggest that participating in roller derby, an aggressive collision sport, is associated with accelerated brain aging, even in the absence of any diagnosed injury. Different patterns of structure–function (de)coupling have been observed in patients with diagnosed neurodegenerative disease [89,90] and after acquired brain injury [91] relative to healthy controls. To the authors’ knowledge, this is the first neuroimaging study to report on the brain organization of post-collegiate female contact and collision sport athletes and is an important first step in characterizing and understanding the risks these sports carry for female brain aging. However, further research is necessary to clarify whether these patterns are consistent across different sports and to what degree they may be mitigated or accelerated by other lifestyle factors. These future investigations are clearly warranted, as policy makers, clinicians, and researchers alike advocate the benefits of sport participation for promoting physical and mental health without a clear understanding of the potential risks for brain aging.

## 5. Conclusions

We compared age-related differences in brain structural and functional networks between adult female roller derby athletes and females not engaged in collision sports. Group differences were observed in patterns of age-related structure-function (de)coupling in the default and frontoparietal networks. These patterns suggest that collision sport participation is associated with faster brain functional dynamics in networks which are typically characterized by neuronal firing on slower timescales. Future studies comparing multimodal imaging and behavior in diverse athlete populations will be necessary to fully understand the risks of collision sports, and head mechanical loading more broadly, for brain aging.

## Figures and Tables

**Figure 1 brainsci-12-00022-f001:**
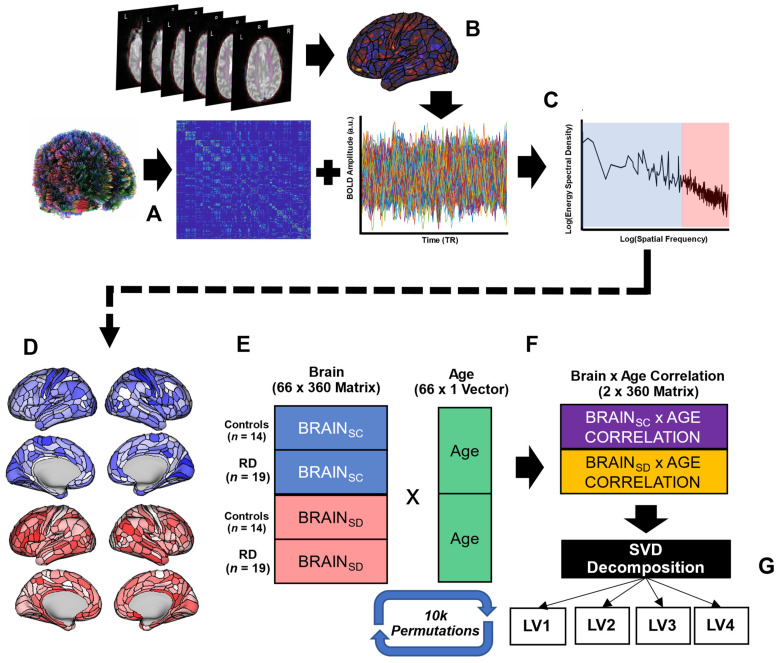
Whole-brain fiber tracking was performed on preprocessed diffusion data to generate a structural connectome as the weighted adjacency matrix (360 × 360) (**A**). Preprocessed resting-state BOLD data were mapped to individual surface reconstructions and averaged within each of 360 cortical areas to represent the graph signal (**B**). Eigen decomposition of the structural connectome was used in a graph Fourier transform of the graph signals into their constituent components such that coupling (S_C_, blue) is observed in the high-energy signals at low spatial frequencies and decoupling (S_D_, red) is observed in the low-energy signals at the highest spatial frequencies (**C**). The norm of high and low energies across TRs reveals the degree of S_C_ (top, blue) and S_D_ (bottom, red) for each cortical area (exemplar control subject) (**D**). In the PLS analysis, each participant is represented by a 1 × 360 vector representing S_C_ (blue) and a 1 × 360 vector representing S_D_ (red) (**E**) and the resulting sample brain matrix (66 × 360) is multiplied by an age vector (66 × 1) to generate correlation matrices (2 × 360) (**F**). These correlation matrices undergo singular value decomposition, revealing 4 latent variables for which *p*-values are generated through permutations (shuffling) of the original data (**G**).

**Figure 2 brainsci-12-00022-f002:**
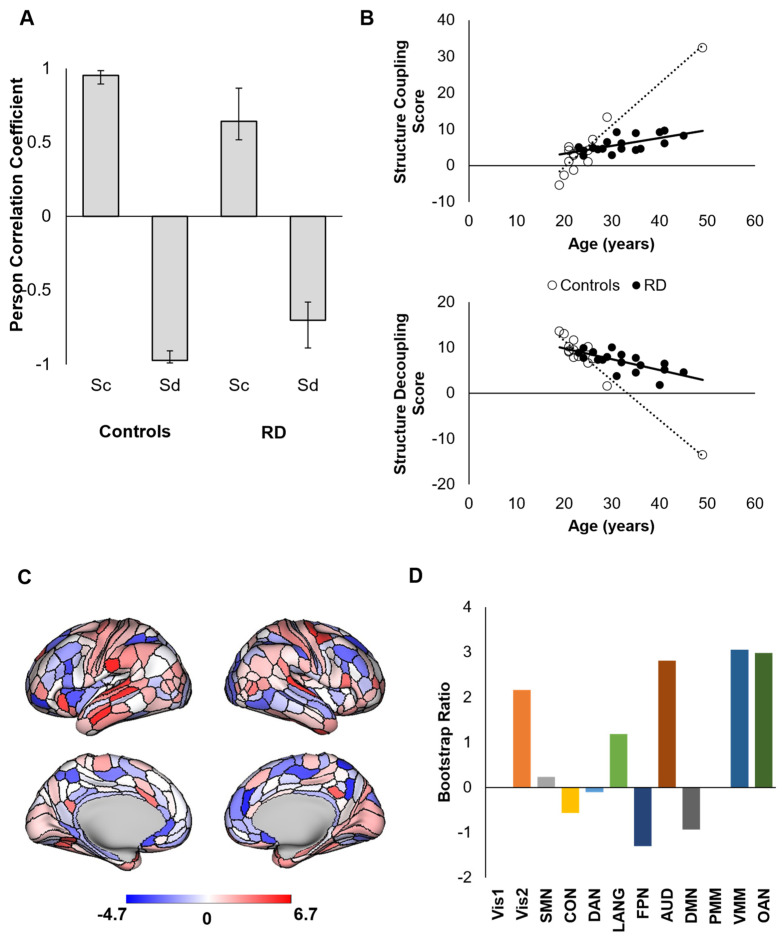
First latent variable from a decomposition of (de)coupling–age correlations between groups (LV1) represented as correlations at the group level (**A**) and individual level (**B**). Positive areal bootstrap ratios contributed reliably to the observed pattern and negative areal bootstrap ratios contributed to the opposite pattern (i.e., greater *decoupling* with age in controls) (**C**). Bootstrap ratios were averaged within networks to reveal the contribution of each network to the observed (de)coupling–age relationships (**D**).

**Figure 3 brainsci-12-00022-f003:**
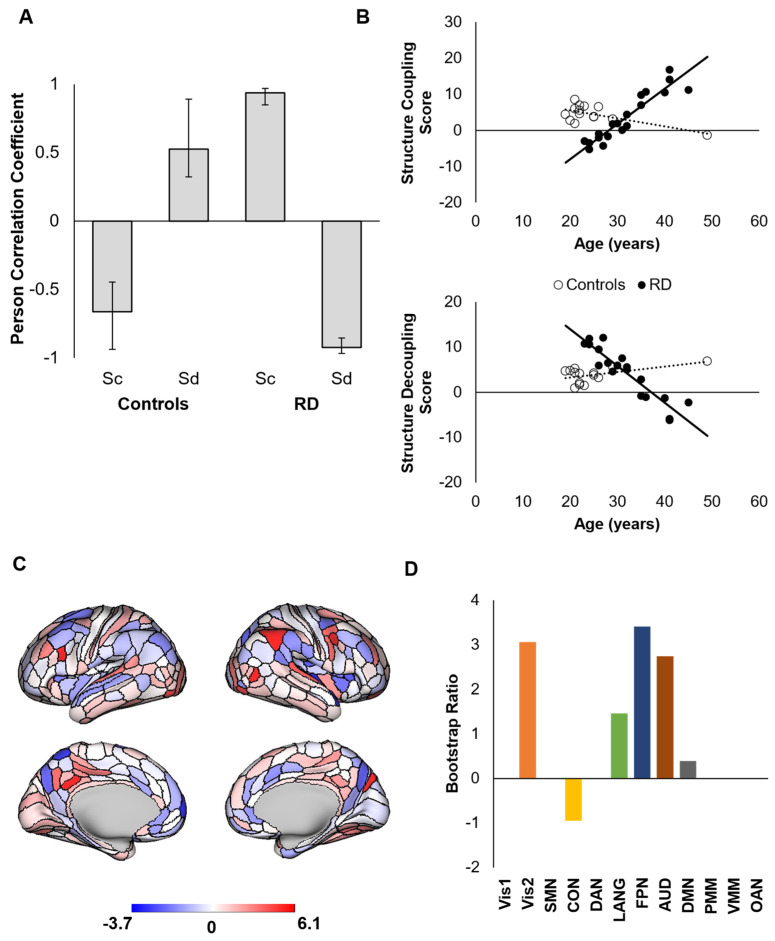
Second latent variable from a decomposition of (de)coupling–age correlations between groups (LV2) represented as correlations at the group level (**A**) and individual level (**B**). Positive areal bootstrap ratios contributed reliably to the observed pattern and negative areal bootstrap ratios contributed to the opposite pattern (i.e., greater *decoupling* with age in RD athletes) (**C**). Bootstrap ratios were averaged within networks to reveal the contribution of each network to the observed (de)coupling–age relationships (**D**).

**Table 1 brainsci-12-00022-t001:** Athlete ages listed in ascending order for both roller derby athletes (left) and controls (right). Head movement expressed as mean framewise displacement (FD). The sports that each control subject reported participating in prior to the beginning of the study or at the time of the study (as denoted by an asterisk *).

Roller Derby (*n* = 19)		Controls (*n* = 14)		
Age (Years)	Mean FD (mm)	Age (Years)	Mean FD (mm)	Sport History
23	0.191	19	0.069	Track, Soccer, Volleyball
24	0.059	20	0.054	Volleyball, Track
24	0.054	21	0.065	Gymnastics
26	0.146	21	0.062	Badminton, Swimming, Tennis
26	0.118	21	0.055	Tennis, Taekwondo, Soccer
27	0.159	22	0.045	Soccer
28	0.061	22	0.109	Volleyball *, Tennis
29	0.062	22	0.066	Track, Volleyball *
30	0.476	23	0.07	Cheerleading, Volleyball, Track
31	0.066	25	0.062	Dance, Competitive Cheer
32	0.095	25	0.138	No
32	0.062	26	0.047	Basketball
35	0.073	29	0.065	Tennis, Lacrosse, Softball, Cheerleading
35	0.072	49	0.069	Soccer, Field Hockey, Basketball, Lacrosse, Softball
36	0.163			
40	0.137			
41	0.076			
41	0.077			
45	0.064			

## Data Availability

The data presented in this study are available on request from the corresponding author. The data are not publicly available because they contain information that could compromise the privacy of research participants.

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
