# Peer review of "Age-Related Trajectories of Brain Structure–Function Coupling in Female Roller Derby Athletes"

_brainsci, 2021, doi:10.3390/brainsci12010022_

Round 1

Reviewer 1 Report

Monroe et al. investigated age-related trajectories of brain structure-function coupling in female roller derby athletes, reporting significant associations between network organization and age. Although the topic is interesting and this study might provide further evidence about the relationship between contact/collision sports and pathophysiological brain mechanisms, there are some issues that reduce my enthusiasm.

My main concern is that the authors investigated age-trajectories in two groups with different ages. This may introduce some biases in the relationship, since it is known that some brain markers (e.g., brain density, white matter integrity, functional connectivity) might show a U-shape relationship (see Luo et al., 2019 HBM). As shown in Figure 1B and 2B, possible U-shape relationships in the whole dataset (controls+athletes) might be observed (after exclusion of outliers). Even if authors are aware that groups were not matched for age, this represents the most important limitation, especially in the framework of this study aimed at investigating aging trajectories brain-couplings.

In first the introduction authors stated that “in part because most neuroimaging studies have been designed to characterize the natural history of concussion over days and weeks in male collegiate athletes [16–19]. This means that the effects of contact and collision sport competition on the female brain are almost entirely unexplored.” Here, I get the point of the authors that wanted to stress that most previous studies were based on male individuals, but from this sentence it might be assumed also that previous studies highlighted brain changes within few weeks from collision, which is in contrast with the emerging literature suggesting a link between collision sports and neurodegeneration occurring later in life. See for instance the Mackay’s seminal work published in the New Engl J Med (2019). Moreover, I would introduce the link between neurodegenerative patterns and collision sports, which is largely missing in the manuscript.

In the second part of the introduction, authors introduced brain functional gradients and the progressive “decoupling” with brain structure from unimodal to transmodal cortices, questioning whether different lifestyles and behaviors can affect this relationship. This statement came out of the blue and it seems unrelated with the main topic of the paper, “Since the purpose of this study was to characterize changes in brain connectivity with aging in female collision sport athletes…”. Moreover, as introduced it is unclear why the main outcome of this study regards the relationship between age and brain organization, but above all why investigating this relationship in a sample of young active athletes. Based on the range of the athletes, authors can focus only in a very limited age of 20 years (23-45). I believe that the paper would benefit from a clearer definition of the aim.

It is unclear whether the HCP dataset includes female individuals without a story of collision sports. If so, please specify in the text. If this information is not available, how authors can be sure that this dataset can serve as control?

Is the fMRI sequence 3.3mm isovoxels? Is fsLR space the native surface or fsaverage space? Please, specify better these parameters.  

At page 4 CABNP appears for the first time. I guess it is the acronym for Cole-Anticevic Brain Network parcellation, but you should specify for readers not familiar with this data. The same sentence presents a typo: each run is reported twice.

A figure summarizing the step analysis would improve the readability of the methodology.

4/14 subjects in the control groups played soccer. There is an emerging literature about the relationship between soccer and dementia risk (see for example Asken et al., 2021 JAMA Neurol). This literature highlighted the role of concussions in soccer. Authors should spent few words on this, commenting also possible biases in the control group.

Reviewer 2 Report

This study revealed the trajectory of structure-function coupling in RD athletes and normal controls. While the topic is interesting, I do have some concerns and comments of the results.

Major concerns:

  1. The motion artifact is unknown. Please include mean FD value into Table 1 for each subjects.  High motion subjects should be excluded.
  2. The calculation of structure-function coupling relies on the intensity of each BOLD volumes. The preprocessing steps of RD/Control dataset  and HCP dataset are different, resulting in different preprocessing results. If you want to include HCP as a reference, a strict way is to use identical pipeline to process all the data.
  3. The PLS model. Please evaluate if mean FD show statistical difference between groups. If so, please add mean FD as covariates.

Minor concerns:

  1. Table 1 is hard to understand correctly. It looks like controls have sport history.
  2. Please add a figure to help illustrate the structure-function coupling.

Round 2

Reviewer 1 Report

All my comments have been addressed

Reviewer 2 Report

The authors have resolved my questions. And I don't have further comments for this manuscript.